# Depressive Symptoms and Gut Microbiota after Bowel Preparation and Colonoscopy: A Pre–Post Intervention Study

**DOI:** 10.3390/microorganisms12101960

**Published:** 2024-09-27

**Authors:** Amelia J. McGuinness, Martin O’Hely, Douglas Stupart, David Watters, Samantha L. Dawson, Christopher Hair, Michael Berk, Mohammadreza Mohebbi, Amy Loughman, Glenn Guest, Felice N. Jacka

**Affiliations:** 1The Institute for Mental and Physical Health and Clinical Translation (IMPACT), School of Medicine and Barwon Health, Deakin University, Geelong, VIC 3220, Australia; a.mcguinness@deakin.edu.au (A.J.M.); martin.ohely@deakin.edu.au (M.O.); samantha.dawson1@deakin.edu.au (S.L.D.); michael.berk@deakin.edu.au (M.B.); amy@thoughtfulspaces.com.au (A.L.); 2Murdoch Children’s Research Institute, Parkville, VIC 3052, Australia; 3School of Medicine, Deakin University, Geelong, VIC 3220, Australia; douglas.stupart@deakin.edu.au (D.S.); david.watters@deakin.edu.au (D.W.); chris@auroragastro.com.au (C.H.); glenn.guest@deakin.edu.au (G.G.); 4Department of Surgery, University Hospital Geelong, Barwon Health, Geelong, VIC 3220, Australia; 5Department of Gastroenterology, Epworth Hospital, Waurn Ponds, VIC 3216, Australia; 6Orygen, Centre for Youth Mental Health, Florey Institute for Neuroscience and Mental Health and the Department of Psychiatry, The University of Melbourne, Melbourne, VIC 3052, Australia; 7Biostatistics Unit, Faculty of Health, Deakin University, Burwood, VIC 3125, Australia; m.mohebbi@deakin.edu.au; 8Centre for Adolescent Health, Murdoch Children’s Research Institute, Melbourne, VIC 3052, Australia; 9College of Public Health, Medical & Veterinary Sciences, James Cook University, Townsville, QLD 4814, Australia

**Keywords:** gut microbiota, colonoscopy, bowel preparation, depression, microbiota–gut–brain axis

## Abstract

Mechanical bowel preparation (MBP) is essential for visualisation of the colon during colonoscopy. Previous studies have identified changes in gut microbiota composition after MBP and colonoscopy. Considering the gut microbiota is increasingly implicated in psychiatry, we explored the potential impact of this intervention on mood and the microbiota–gut–brain axis. We conducted a pre–post intervention study in adults, with timepoints of one week before and one month after MBP and colonoscopy. Our primary outcome was change in average Hospital Anxiety and Depression Scale depression sub-scores. We examined changes in average anxiety, stress, and quality of life scores and gut microbiota composition using 16S rRNA sequencing. We further explored associations between changes in depressive symptoms and gut microbiota and conducted post hoc analyses to explore potential effect modifiers. Average depressive symptom scores decreased one month post-procedure compared to baseline (n = 59; adjusted β = −0.64; 95%CI: −1.18, −0.11). Irritable bowel syndrome (IBS) appeared to moderate this relationship (β = 1.78; 95%CI: 0.292, 3.26); depressive symptoms increased in those with, and decreased in those without, IBS. Reduced alpha diversity, modest effects on beta-diversity, and increases in health-associated genera were observed one month post-procedure. Increases in the CLR-transformed abundances of *Ruminococcaceae UCG-009* were associated with improvements in depressive symptoms. There is preliminary evidence of a potential mental health effect of MBP and colonoscopy, particularly for those with IBS, which may be associated with changes to the gut microbiota. Further research is required to confirm these findings and their clinical relevance.

## 1. Introduction

The microbiome–gut–brain axis has emerged as a contributing factor to the aetiology and pathophysiology of depression [1,2]. Preclinical evidence suggests that many physiological pathways influenced by the gut microbiota are also implicated in depression [3]. In humans, differences in gut microbiota composition have been associated with depression [4,5], and interventions that modulate the gut microbiota, such as diet [6,7,8] and probiotic supplementation [9], can alleviate depressive symptoms. Manipulation of the gut microbiota may further our understanding of how the gut microbiota influences, and can be targeted to treat, depressive disorders. Reciprocally, understanding how common interventions affect the gut microbiota may have important clinical and pathophysiological implications.

Mechanical bowel preparations (MBPs) are consumed prior to colonoscopy for visualisation of the colon and the detection of abnormalities during colonoscopy [10,11]. Previous research has shown an impact of MBP on the gut microbiome. Indeed, osmotic laxatives have strong associations with gut microbiota composition and function, with large effect sizes [12]. Previous intervention studies that have examined the impact of MBP and colonoscopy (hereafter referred to as the ‘procedure’) on the gut microbiota have reported mixed results. Whilst decreases in the number and distribution (i.e., alpha diversity) and relative abundances of bacterial taxa have been observed immediately after MBP (before colonoscopy) [13,14], very minimal or no changes have also been described [15,16]. Differences in gut microbiota composition have been reported one week [14,15,17,18,19] to one month [13,17,20,21] post-procedure compared to baseline; however, in other studies, there were no statistically significant differences at these follow-up time points [17,19,20]. Few studies have measured changes in gut microbiota composition beyond one month post-procedure, and those that have provided limited evidence of ongoing changes [18,22].

Whilst studies have aimed to characterise changes in gut microbiota composition, no studies have investigated changes in depressive symptoms after MBP and colonoscopy or explored the potential associations between changes in gut microbiota composition and changes in depressive symptoms. Considering the millions of colonoscopies conducted annually [23,24], there is a need to better understand the potential impact of this procedure on the microbiome–gut–brain axis.

The aim of this study was to investigate changes in self-reported depressive symptoms one week before versus one month after MBP and colonoscopy. We also aimed to explore changes in gut microbiota composition and functional potential post-procedure and identify any associations with changes in depressive symptoms.

## 2. Materials and Methods

### 2.1. Study Design

The data used for this manuscript were derived from the Micro-Scope study, which used a pre–post intervention study design to investigate changes in depressive symptoms and gut microbiota composition after MBP and colonoscopy. Data are reported in accordance with the Strengthening the Reporting of Observational Studies in Epidemiology (STROBE) Statement [25], and the Strengthening the Organising and Reporting of Microbiome Studies (STORMS) checklist [26]. This study had ethical approval from the Barwon Health (#15-129), Epworth Healthcare (#EH2016-146), and Deakin University (#2016-391) Human Research Ethics Committees. Microbial data were not uploaded to an online public repository as consent for data sharing was not obtained. This study was pre-registered on the Open Science Framework (OSF): https://osf.io/fv7xd/ (accessed on 17 September 2024).

### 2.2. Participants and Setting

Participants were adults referred for colonoscopy between May 2017 and November 2018 at University Hospital Geelong, Australia. Colonoscopies were performed at University Hospital or Epworth Hospital, Geelong, Australia. Recruitment occurred at the time of their initial outpatient consultation with the General Surgery or Gastroenterology services. Any adults referred for colonoscopy during the study time frame were considered for recruitment. Exclusion criteria were individuals who were highly dependent on medical care or unable to give informed consent (e.g., language barriers, significant intellectual or cognitive disability). There were no exclusions regarding antibiotic use. Eligible participants were directed to a member of the research team to discuss the study further and provide informed consent. Those diagnosed with cancer post-colonoscopy were withdrawn and did not provide follow-up data.

### 2.3. Data Collection

One week pre-procedure, participants completed paper-based questionnaires and collected a fresh faecal sample in a sterile collection jar at home. The sample was stored in their freezer (−20 °C) for approximately one week until being transported on ice to the research team on the day of colonoscopy, when it was transferred to a −80 °C freezer for storage until DNA extraction. Luminal aspirates were collected during colonoscopy to demonstrate the potential immediate changes in alpha diversity associated with the procedure. Faecal residue from the colon was aspirated into a sterile collection jar (flushed with saline for collection if necessary), placed on ice, then transferred to a −80 °C environment. One month post-procedure, participants completed another set of questionnaires and collected a final faecal sample at home as previously described.

### 2.4. Intervention

Bowel preparation was prescribed and carried out as per normal advice and practice for the colonoscopy service. Participants were instructed to commence a low-fibre (“white”) diet two days before their colonoscopy, and then fast for 12–24 h prior to their procedure and consume a sodium picosulfate-based bowel preparation product in three separate doses. Adequacy of bowel preparation was reported during colonoscopy by the endoscopist using a modified overall Boston Bowel Preparation Scale score [27].

### 2.5. Outcomes

Depressive symptoms (primary outcome) were measured using the depression sub-score of the Hospital Anxiety and Depression Scale (HADS) [28]. All other outcomes were considered exploratory. The severity of depressive symptoms was measured using the Patient Health Questionnaire-9 (PHQ-9) [29]; anxiety symptoms using the anxiety sub-score of the HADS [28]; total, psychosocial, and physical quality of life using the Assessment of Quality of Life-8 (AQOL-8D) [30]; and stress using the Perceived Stress Scale (PSS) [31]. Gut microbiota alpha diversity was measured using the Shannon index—a within-sample index of both richness and evenness—and the number of observed amplicon sequencing variants (ASVs) at the genus level. Beta-diversity was measured using the Aitchison distance and differential abundances of genera were determined using centred-log ratio (CLR)-transformed count abundance data.

### 2.6. Covariates

Age at the time of recruitment and sex (male/female/other) were obtained from medical records. A triage nurse collected participant height and weight at their initial outpatient consultation, which were used to calculate body mass index (BMI; kg/m^2^). Participants self-reported their residential postcode and suburb, which were used to calculate socioeconomic status using an area-based measure called the Index of Relative Socio-economic Advantage and Disadvantage (IRSAD) [32]. Each suburb has an IRSAD classification ranging from 1 to 10, where a lower IRSAD score suggests greater disadvantage. Smoking status, lifetime history of medical conditions (including depression), and current medication use were self-reported. Diet quality was measured using the Simple Dietary Questionnaire [33], where the total score (out of 100) rated dietary adherence to the Australian Dietary Guidelines, with higher scores representing greater compliance [34]. The ROME III Diagnostic Questionnaire for Adult Functional Gastrointestinal Disorders was used to determine if participants met irritable bowel syndrome (IBS) criteria [35]. Colonoscopy indication and outcomes were obtained from medical records.

### 2.7. DNA Extraction

Microbial DNA extraction from stool was performed using the commercial QIAamp Fast DNA Stool Mini Kit (QIAGEN, Hilden, Germany) as per manufacturer instructions, with an additional mechanical lysis step using PowerBead tubes (QIAGEN, Hilden, Germany). Extracted DNA was stored at −80 °C until it was couriered on dry ice to the Australian Genomic Research Facility (AGRF) for sequencing.

### 2.8. Sequencing and Annotation

Sequencing of the 16S rRNA gene sequence was performed using the Illumina MiSeq (Illumina, San Diego, CA, USA) platform. The V1–V3 hypervariable region of the 16S rRNA gene was amplified by polymerase chain reaction using 27F (AGAGTTTGATCMTGGCTCAG) and 519R (GWATTACCGCGGCKGCTG) primers with a read length of 300 base pairs. Diversity profiling analysis was performed with QIIME 2 2019.7 [36]. The demultiplexed raw reads were primer trimmed and quality filtered using the cutadapt plugin followed by denoising with DADA2 (via q2-dada2) [37]. Taxonomy was assigned to amplicon sequence variants (ASVs) using the q2-feature-classifier [38] classify-sklearn naïve Bayes taxonomy classifier. Taxonomy was assigned using the SILVA (v.132) database.

### 2.9. Pre-Processing

Pre-processing and filtering was performed as described by Callahan et al. [39]. Zero-count bacterial features and non-bacterial taxa were removed prior to calculating the alpha diversity metrics, Shannon index, and observed ASVs. Additional filtering removed low-prevalence taxa (those present in less than 5% of samples), and data were centred-log ratio transformed to calculate Aitchison distances (i.e., beta-diversity) and for differential abundance testing. Functional potential was predicted using the Phylogenetic Investigation of Communities by Reconstruction of Unobserved States (PICRUSTt2) [40], which maps to MetaCyc pathways.

### 2.10. Power Calculation

A sample size of 59 has more than 80% power to detect mean differences on a paired before–after comparison with a HADS score of 1.5 or greater, which has been considered clinically important [41]. We assumed a standard deviation of 3.7 for the differences [42], and a significance level of 0.05 using a two-sided paired *t*-test was used in the power calculation.

### 2.11. Statistical Analyses

Statistical analyses were performed using the RStudio [43] 4.3.1 environment. See the Appendix A for the R packages used. All models included age, sex, BMI, diet quality, and IBS at baseline as covariates, as these have previously been associated with depression. Multiple imputation with predictive mean matching was performed for missing covariate data using the mice [44] package, including age, sex, and BMI as auxiliary variables for precision.

The primary outcome was complete case analysis of the change in average HADS depressive symptom scores one week pre- versus one month post-procedure using generalised estimating equations (GEEs) via the geepack [45] and rstatsToolkit [46] packages, assuming a Gaussian distribution with an AR(1) correlation structure to account for within-participant autocorrelation. Changes in average depressive symptom severity, anxiety symptoms, quality of life, perceived stress scores, Shannon index, observed genera, and the Shannon index of MetaCyc pathways were also determined using GEEs as above. Robust standard error estimates were reported for all GEE models. Beta-diversity pre- and post-intervention was plotted using Principal Component Analysis (PCA) of Aitchison distances. Changes in beta-diversity metrics across time points were calculated using pairwise permutational ANOVA (adonis2) with 999 permutations, stratified by participant ID to consider the paired nature of the data, via the vegan [47] package. Differential abundance analyses at the genus level and of functional MetaCyc pathways were calculated using linear mixed models in the Maaslin2 [48] package, with minimum abundance and prevalence set to zero, time point and covariates as fixed effects, participant as the random effect, and a CLR transformation.

Exploratory linear regression models were used to examine associations between the changes in average HADS depression scores with changes in bacterial genera and alpha diversity. A post hoc sensitivity analysis was performed, whereby all participants with a baseline sample were included in a modified intention-to-treat (mITT) analysis with missing follow-up data imputed using predictive mean matching, with age, sex, BMI, marital status, employment status, socioeconomic decile, diet quality, IBS, and bowel preparation adequacy used as auxiliary variables.

For differential abundance and exploratory analyses, the Benjamani–Hochberg procedure was applied to control the false discovery rate (FDR), with taxa below an FDR of 0.05 reported in the results.

## 3. Results

### 3.1. Recruitment

We enrolled 136 participants who provided informed consent at the time of their outpatient appointment. Eighty-six participants were successfully contacted and provided baseline data. Reasons for non-participation in the study are outlined in Figure 1. Of these 86 participants, 5 participants were excluded from analyses due to inadequate MBP (rated as ‘poor’ by their endoscopist) and 2 were excluded due to cancer diagnosis post-procedure, with 79 participants remaining for modified intention-to-treat analysis. Of these, a further 20 participants were lost to follow up between the baseline and follow-up time points, with reasons outlined in Figure 1. Therefore, 59 participants were included in primary analyses (Figure 1). In addition, two participants did not have intra-colonoscopy samples collected, and 3 intra-colonoscopy samples were of too low a biomass to yield sufficient DNA, leaving 56 luminal samples for analysis.

### 3.2. Participant Characteristics

Participant characteristics are presented in Table 1. Participants had a mean age of 58.5 years, an average BMI of 29.7, and an almost equal distribution of sex (54% female) and socioeconomic advantage (56% IRSAD > 5). Most were taking at least one medication (92%) and had poor average diet quality (46.8/100 SDQ points), with 12% being current smokers, 22% meeting diagnostic criteria for IBS, and 22% self-reporting a lifetime history of depression. No participants self-reported antibiotic use one week pre-procedure. Procedural characteristics are presented in Appendix A. Faecal samples were collected (on average) 7 days pre-procedure and 33 days post-procedure. The MBP adequacy of most participants was rated as ‘good’ (but not excellent) by the endoscopists (63%), and polyps were the most common finding during colonoscopy (63%).

### 3.3. Change in Mental Health Symptoms

There were decreases in average HADS and PHQ-9 depression scores one month post-procedure versus baseline (Table 2). There were increases in the average total, psychosocial, and physical quality of life scores but no statistical evidence of changed average HADS anxiety scores or perceived stress scores (Table 2). Sensitivity analyses were similar; however, the reduction in perceived stress scores became statistically significant (Table 2).

### 3.4. Changes in Gut Microbiota Composition

Luminal samples collected during colonoscopy had lower average alpha diversity compared to faecal samples collected pre- and post-procedure (Figure 2).

Statistical testing was performed using only the faecal samples collected one week pre- and one month post-procedure. The average number of observed genera decreased one month post-procedure versus baseline; however, there was no statistically significant change in the average Shannon index (Table 2). There was a change in overall beta diversity one month post-procedure (*p* < 0.001); however, the time point only explained 0.4% of the variance after adjusting for baseline covariates (Appendix A).

There were 22 differentially abundant genera one month post-procedure compared to baseline after adjusting for baseline covariates and multiple comparisons (*q* < 0.05) (Table 3). Only an unidentified *Lachnospiraceae* genus was decreased one month post-procedure compared to baseline, whereas the other 21 genera were increased post-procedure, and included the following annotated/identified taxa: *Cutibacterium*, *Prevotella 9*, *Megamonas*, *Ruminococcaceae UCG-009*, *Oxalobacter*, *Lactonifactor*, *Ruminococcaceae CAG352*, *Lachnospiraceae GCA900066575*, *Eubacterium, Gordonibacter*, *Solobacterium*, *Rikenellaceae RC9 group*, *Lachnospiraceae UCG-010*, and *Megasphaera* (Table 3).

There were no statistically significant changes in the average alpha diversity (Shannon index) of functional MetaCyc pathways (Table 2). Differential abundance analyses identified increases in ten MetaCyc pathways one month post-procedure versus baseline after adjustment for baseline covariates and multiple comparisons (*q* < 0.05) (Table 4).

### 3.5. Exploratory Post Hoc Analyses

We examined whether IBS at baseline moderated the observed decrease in average HADS depression scores post-procedure. We observed a statistically significant two-way interaction between IBS and time point (β = 1.78; 95%CI: 0.29, 3.26). Those without IBS appeared to have reductions in HADS depression scores, whereas those with IBS appeared to have increased HADS depression scores one month post-procedure compared to baseline (Figure 3). Additionally, we examined whether BMI at baseline moderated the observed decrease in average HADS depression scores post-procedure. We observed a statistically significant two-way interaction between baseline BMI (above/below 30) and time point (β = −1.15; 95%CI: −2.18, −0.13). Those with a BMI equal to or greater than 30 appeared to have reductions in HADS depression scores, whereas those with a BMI below 30 appeared to have minimal changes in HADS depression scores one month post-procedure compared to baseline (Figure 3).

We also examined whether colonoscopy outcome moderated the observed decrease in average HADS depression scores post-procedure. Due to our small sample size and some participants receiving more than one colonoscopy outcome, we dichotomised participants into those with (80%) and without abnormalities identified. We did not observe evidence of a statistically significant two-way interaction between colonoscopy outcome and time point (β = −0.69; 95%CI: −2.11, 0.72). Additionally, we explored whether the use of antidepressants moderated the observed decrease in HADS depression scores post-procedure; however, we did not observe evidence of a statistically significant two-way interaction between antidepressant use (yes/no) and time point (β = 0.46; 95%CI: −1.02, 1.93).

### 3.6. Post Hoc Analyses

We explored associations between the changes in gut microbiota composition and changes in average HADS depression scores. There were eight genera whose change was associated with changes in depressive symptoms after adjusting for covariates (*p* < 0.05) (Table 5). An inverse association between *Ruminococcaceae UCG-009* and the change in average HADS depression scores remained significant after adjusting for multiple comparisons (*q* = 0.028) (Table 5).

## 4. Discussion

This is the first study, to our knowledge, to investigate the potential impact of colonoscopy and MBP on depressive symptoms and to associate changes in depressive symptoms with changes to gut microbiota composition. We observed a decrease in average depressive scores and increases in average quality of life scores one month post-procedure compared to baseline, but there was little evidence to support decreased average perceived stress or anxiety scores. The changes in average depression scores appeared to be moderated by IBS, whereby those with IBS experienced worsening of their depressive symptoms, and those without IBS experienced improvements. Compared to baseline, we also observed reduced gut microbiota alpha-diversity, modest effects on beta-diversity, and differentially abundant genera and functional pathways. Finally, we observed that increases in the genus *Ruminococcaceae UCG-009* were associated with decreases (i.e., improvements) in depressive symptoms.

The finding that depressive symptoms decreased one month post-procedure has not been previously explored. This finding is somewhat concordant with studies of other microbiome-modulating interventions that report improvements in depressive symptoms, including probiotics [9], faecal microbiota transplant [49,50], and diet [6,8]. However, it is plausible that our participants experienced improvements in their depressive symptoms due to no major adverse findings (i.e., cancer) during colonoscopy, although we did not observe any changes in stress and anxiety measures, which might be expected if this were the explanation. Moreover, levels of depressive symptoms at baseline were low, and the average reduction observed, although statistically significant, was very small.

We observed a two-way interaction suggesting that those with IBS at baseline had worsening of their depressive symptoms post-procedure, whereas those without IBS experienced improvements. Interpretation of this finding is limited by the small number of participants with IBS. However, it is possible that their increase in depression symptoms could have been due to an aggravation of their IBS symptoms post-procedure. Worsening of gastrointestinal symptoms occurs in ~20% of patients after colonoscopy [51], and those with IBS are more likely to experience prolonged post-procedural abdominal pain [52]. Worsening depressive symptoms could also be due to a lack of findings to adequately explain their IBS symptoms. A previous study of IBS patients showed no improvement in reassurance or health-related quality of life in those that received a negative colonoscopy result [53]. Additionally, a study observed that those with IBS were more likely to experience lower satisfaction, and higher burden, and embarrassment immediately after colonoscopy [54], and that these perceptions were worse six weeks post-colonoscopy [54]. Whether individuals with IBS represent an at-risk group that require additional disease management after colonoscopy may be an important focus of further research.

We also observed two-way interaction suggesting that those with an obese-range BMI (equal to or greater than 30) experienced greater improvement in their depressive symptom scores compared to those with a BMI below 30, who experienced minimal change. Preclinical evidence has provided strong evidence of the importance of the gut microbiota in obesity [55,56]. Although there is contention regarding the specific bacterial signatures, evidence consistently suggests that obesity in humans is associated with different gut microbiota compositions [57,58]. The gut microbiota are thought to contribute to obesity via increasing the energy extracted from food sources, influencing metabolic processes associated with fat accumulation, and contributing to the inflammation associated with obesity and metabolic syndrome via interactions with the immune system [59]. Obesity is also strongly associated with depression, and both share many overlapping biological mechanisms including changes to the immune system and inflammation, neuroendocrine dysregulation, and even structural and functional brain alterations, all of which are also associated with the gut microbiota [60]. Whilst our study was not adequately powered to explore this hypothesis, whether individuals with a high BMI experience a greater ‘resetting’ of their gut microbiota in response to MBP with downstream beneficial impacts on mental health warrants further investigation.

Many bacterial genera that were higher in abundance one month post-procedure compared to baseline have potential functionalities that may be advantageous for health. We observed increases in genera with the capacity to metabolise dietary polyphenols into metabolites that confer health benefits, including *Gordonibacter* [61,62] and *Lactonifactor* [63]. Low hippurate, produced by the bacterial metabolism of polyphenols, has been causally related to depression [64]. There were also increases in bacteria with butyrate-producing capacity, including *Eubacterium*, *Solobacterium*, and *Megasphaera*; the health benefits of butyrate have been reviewed extensively [65]. Thus, it is possible that the flushing out of gut bacteria by this procedure may ‘reset’ the gut microbiota, resulting in a composition better able to metabolise foods and harness benefits from their diet. We also observed higher abundances of ten MetaCyc pathways, reflecting a range of different processes relating to nutrient metabolism, one month post-procedure. However, as this study used 16S rRNA sequencing, we could only measure inferred functionality using PICRUSt2; therefore, the biological relevance is unclear and requires exploration with shotgun metagenomic sequencing.

We observed that as *Ruminococcaceae UCG009* increased, depressive symptoms improved. Our previous systematic review found that *Ruminococcaceae* was consistently lower in individuals with mental disorders compared to healthy controls [4]. A large association study found *Ruminococcaceae UCG002*, *UCG003*, and *UCG005* to be negatively associated with depressive symptoms [66]. The *Ruminococcaceae* family is largely considered beneficial to health, including many butyrate-producing genera such as *Ruminococcus*, *Faecalibacterium*, *Caproiciproducens*, *Agathobaculum*, *Butyricicoccus*, and *Gemmiger*. Future studies employing whole-genome metagenomic sequencing with greater resolution at the species and strain level, or those that specifically target changes in the abundance of *Ruminococcaceae* taxa, may afford further insight into how these bacteria change after intervention.

Our study has notable limitations. We used 16S rRNA gene sequencing, which is subject to variability in sequencing depth and has only genus-level resolution. We did not collect biological samples such as blood or urine to investigate changes to bacterial metabolites. Our study had a small sample size, and colonoscopy indications and outcomes were heterogenous, which limited our statistical power and prevented subgroup investigations. As we recruited a convenience sample of adults undergoing colonoscopy at our regional public hospital, our study population was highly heterogenous in terms of their baseline clinical characteristics, and the impact of these differences on baseline gut microbiota composition and re-establishment are unclear. Whilst no participants reported antibiotic use within one week of colonoscopy, the use of antibiotics within one month of colonoscopy was not an exclusion criterion, which may have impacted baseline composition. Stool samples were collected at home and stored in participants’ home freezers for one week prior to transfer to long-term storage, and the impact of the delay in storage at −80 °C is also unclear. Some of the changes observed may be associated with colonoscopy itself rather than MBP, as the use of propofol, a sedative used for colonoscopy in the present study, has been previously found to have antidepressant potential [67]. All participants in this study specifically used a sodium picosulfate-based MBP, and different results may be observed with alternative MBP products. Those lost to follow-up had poorer mental health compared to those that completed the study, and the impact of their missing data, particularly for perceived stress, is unknown. Finally, we only collected comparable faecal samples at two time points, and greater sampling frequency within and beyond one month post-procedure may provide additional information regarding how the gut microbiota re-establishes.

## 5. Conclusions

Our study provides preliminary evidence of a potential impact of MBP and colonoscopy on depressive symptoms that may relate to changes in the gut microbiota. Future research leveraging MBP as a method of gut microbiota modulation may further our understanding of the microbiota–gut–brain axis. To better elucidate any potential mental health impact of MBP itself, clinical trials of this intervention in a population experiencing heightened psychological distress and without colonoscopy are needed. Finally, the clinical implications of the potential differential impact of MBP and colonoscopy in those with and without IBS deserve further exploration.

## Figures and Tables

**Figure 1 microorganisms-12-01960-f001:**
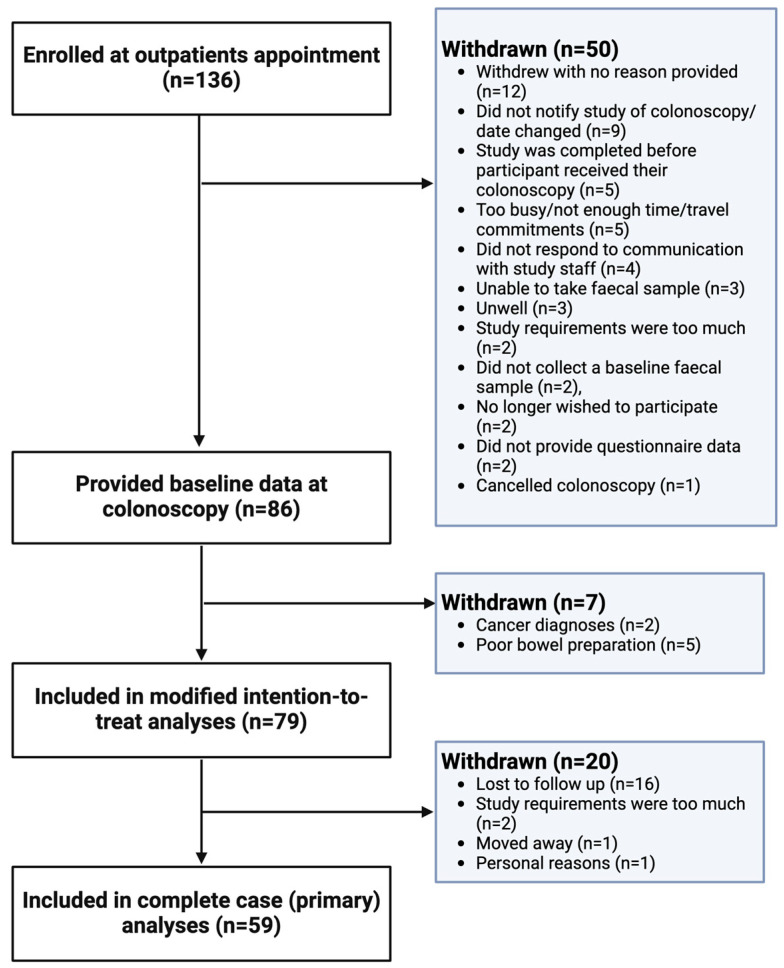
Participant enrolment and participation flow chart.

**Figure 2 microorganisms-12-01960-f002:**
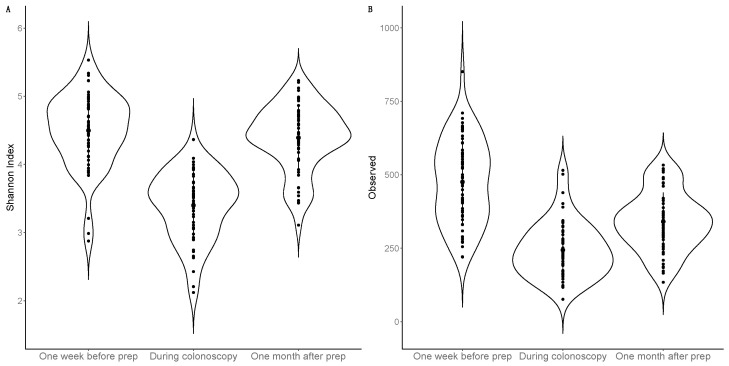
Differences in alpha-diversity metrics of (**A**) Shannon index and (**B**) observed genera one week before bowel preparation and colonoscopy (n = 59, orange), during colonoscopy (n = 56, purple), and one month after bowel preparation and colonoscopy (n = 59, blue).

**Figure 3 microorganisms-12-01960-f003:**
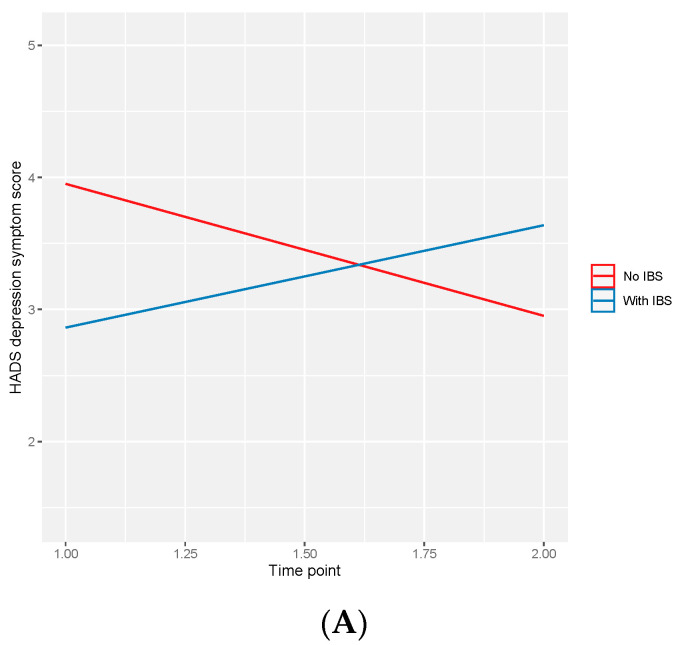
Two-way interaction between average HADS depressive symptom scores and (**A**) irritable bowel syndrome and (**B**) body mass index at baseline. Shaded regions are confidence intervals for the estimated marginal means.

**Table 1 microorganisms-12-01960-t001:** Baseline characteristics of participants included in the present analyses (complete cases) and those lost to follow up.

	Complete Cases (n = 59)	Lost to Follow Up (n = 20)	All Cases (n = 79)
**Demographic**			
Age, years, mean (SD)	58.5 (11.2)	56.7 (15.6)	58.0 (12.4)
Sex, female, n (%)	32 (54%)	13 (65%)	45 (57%)
IRSAD SES decile > 5, n (%)	33 (56%)	10 (50%)	43 (54%)
**Health measures**			
BMI, mean (SD)	29.7 (6.42)	27.8 (6.21)	29.2 (6.38)
Current smoking, yes, n (%)	7 (12%)	6 (30%)	13 (17%)
Diet quality, mean (SD)	44.8 (11.9)	30.2 (12.6)	41.7 (13.4)
Missing, n (%)	0 (0%)	4 (20.0%)	4 (5.1%)
**Medical**			
Self-report depression, yes n (%)	13 (22%)	6 (30%)	19 (24%)
Missing, n (%)	2 (3%)	4 (20%)	6 (7.6%)
IBS, yes n (%)	13 (22%)	6 (30%)	19 (24%)
IBS—constipation predominant	2 (15%)	2 (33%)	4 (21%)
IBS—diarrhoea predominant	6 (46%)	0 (0%)	6 (32%)
IBS—mixed predominant	5 (39%)	4 (67%)	9 (47%)
IBS—undefined predominant	0 (0%)	0 (0%)	0 (0%)
Missing, n (%)	1 (2%)	4 (20%)	5 (6%)
**Medications**			
Any medication, yes n (%)	54 (92%)	18 (90%)	72 (91%)
Antidepressants	11 (19%)	7 (35%)	18 (23%)
Amitriptyline	2 (3.4%)	1 (5.0%)	3 (3.8%)
Desvenlafaxine	1 (1.7%)	1 (5.0%)	2 (2.5%)
Duloxetine and nortriptyline	1 (1.7%)	0 (0%)	1 (1.3%)
Escitalopram	2 (3.4%)	2 (10.0%)	4 (5.1%)
Escitalopram and mirtazapine	1 (1.7%)	0 (0%)	1 (1.3%)
Moclobemide	1 (1.7%)	0 (0%)	1 (1.3%)
Sertraline	3 (5.1%)	1 (5.0%)	4 (5.1%)
Venlafaxine	0 (0%)	1 (5.0%)	1 (1.3%)
Not specified	0 (0%)	1 (5.0%)	1 (1.3%)
Hyperacidity/reflux medication	22 (37%)	6 (30%)	28 (35%)
Aluminium hydroxide–magnesium Hydroxide–simethicone	1 (1.7%)	0 (0%)	1 (1.3%)
Calcium carbonate–magnesium carbonate hydrate	1 (1.7%)	0 (0%)	1 (1.3%)
Calcium carbonate–magnesium Carbonate–magnesium trisilicate	1 (1.7%)	0 (0%)	1 (1.3%)
Esomeprazole	5 (8.5%)	4 (20.0%)	9 (11.4%)
Lansoprazole	0 (0%)	1 (5.0%)	1 (1.3%)
Omeprazole	1 (1.7%)	0 (0%)	1 (1.3%)
Pantoprazole	6 (10.2%)	1 (5.0%)	7 (8.9%)
Pantoprazole and ranitidine	1 (1.7%)	0 (0%)	1 (1.3%)
Rabeprazole sodium	1 (1.7%)	0 (0%)	1 (1.3%)
Rabeprazole sodium and calcium carbonate–magnesium carbonate– magnesium trisilicate	1 (1.7%)	0 (0%)	1 (1.3%)
Ranitidine	3 (5.1%)	0 (0%)	3 (3.8%)
Sodium alginate–sodium bicarbonate–calcium carbonate	1 (1.7%)	0 (0%)	1 (1.3%)
Digestive supplements/cholelitholytics	2 (3%)	0 (0%)	2 (3%)
**Mental health symptoms, mean (SD)**			
HADS depression	3.71 (3.06)	5.45 (4.03)	4.15 (3.39)
HADS anxiety	5.37 (3.74)	7.20 (4.63)	5.84 (4.03)
HADS total	9.08 (6.04)	12.7 (7.98)	9.99 (6.71)
PHQ9 depression severity	3.61 (5.38)	7.20 (7.14)	4.52 (6.03)
Perceived stress	10.7 (7.67)	15.5 (7.14)	11.9 (7.77)
**Quality of Life, mean (SD)**			
Total	0.77 (0.19)	0.64 (0.24)	0.74 (0.21)
Psychosocial	0.48 (0.21)	0.36 (0.22)	0.45 (0.22)
Physical	0.71 (0.17)	0.64 (0.23)	0.69 (0.19)

Note: Age is age at time of recruitment; BMI calculated as weight (kg)/height(m)^2^ at time of recruitment; diet quality calculated from the Simple Dietary Questionnaire based on previous studies [34]. Abbreviations: BMI, body mass index; HADS, Hospital Anxiety Depression Scale; IBS, irritable bowel syndrome; PHQ, Patient Health Questionnaire; SD, standard deviation; SDQ, Simple Dietary Questionnaire.

**Table 2 microorganisms-12-01960-t002:** Changes in mental health symptoms and gut microbiota after bowel preparation and colonoscopy.

			Complete-Case Analysis (n = 59)		Modified ITT Sensitivity Analysis ^a^ (n = 79)	
	Baseline Mean (SD)	Final Mean (SD)	Unadjusted β (95% CI)	Adjusted β (95% CI)	Unadjusted β (95% CI)	Adjusted β (95% CI)
Mental health and quality of life outcomes						
Depressive symptoms	3.71 (3.06)	3.07 (2.83)	−0.64 (−1.18 to −0.11)	−0.67 (−1.19 to −0.15)	−0.80 (−1.33, −0.26)	−0.97 (−1.58, −0.37)
Depressive symptom severity	3.61 (5.38)	2.98 (5.07)	−0.63 (−1.29 to 0.04)	−0.68 (−1.33 to −0.02)	−0.84 (−1.51, −0.17)	−1.41 (−2.33, −0.50)
Anxiety symptoms	5.37 (3.74)	5.27 (4.18)	−0.10 (−0.63 to 0.42)	−0.09 (−0.61 to 0.44)	−0.19 (−0.72, 0.35)	−0.58 (−1.32, 0.16)
Perceived stress	10.7 (7.67)	10.4 (7.30)	−0.32 (−1.53 to 0.89)	−0.39 (−1.61 to 0.82)	−0.57 (−1.76, 0.62)	−1.86 (−3.21, −0.51)
Total quality of life	0.77 (0.19)	0.80 (0.19)	0.02 (0.01 to 0.04)	0.03 (0.01 to 0.04)	0.03 (0.01, 0.04)	0.05 (0.02, 0.09)
Psychosocial quality of life	0.48 (0.21)	0.52 (0.22)	0.04 (0.02 to 0.06)	0.04 (0.02 to 0.06)	0.04 (0.02, 0.06)	0.05 (0.02, 0.09)
Physical quality of life	0.71 (0.17)	0.73 (0.18)	0.02 (0.00 to 0.04)	0.02 (0.00 to 0.05)	0.03 (0.00, 0.05)	0.05 (0.01, 0.09)
Gut microbiota outcomes						
Observed genera	476 (143)	341 (98.8)	−135 (−168 to −102)	−137 (−170 to −105)	−123 (−154, −92.1)	−99.7 (−132, −67.0)
Genus-level Shannon index	4.5 (0.54)	4.39 (0.48)	−0.10 (−0.23 to 0.02)	−0.10 (−0.23 to 0.03)	−0.04 (−0.17, 0.08)	−0.06 (−0.20, 0.08)
MetaCyc group Shannon index	5.17 (0.11)	5.16 (0.12)	−0.01 (−0.04 to 0.02)	−0.01 (−0.04 to 0.02)	−0.01 (−0.04, 0.02)	−0.02 (−0.05, 0.01)

^a^ Post hoc sensitivity analysis including all participants with a baseline sample with missing follow-up data imputed using predictive mean matching, with age, sex, body mass index, marital status, employment status, socioeconomic decile, diet quality, irritable bowel syndrome, and bowel preparation adequacy used as auxiliary variables. Abbreviations: CI, confidence interval; ITT, intention-to-treat; SD, standard deviation.

**Table 3 microorganisms-12-01960-t003:** Adjusted analyses of the change in differential abundance of taxa at the genus level.

Family	Genus	Coefficient (SE)	*p*-Value	*q*-Value
Lower				
*Lachnospiraceae*	*Unidentified*	−0.36 (0.09)	<0.001	<0.001
Higher				
*Propionibacteriaceae*	*Cutibacterium*	0.30 (0.04)	<0.001	<0.001
*Prevotellaceae*	*Prevotella 9*	0.24 (0.07)	0.001	0.001
*Veillonellaceae*	*Megamonas*	0.23 (0.03)	<0.001	<0.001
*Flavobacteriaceae*	*Uncultured*	0.23 (0.03)	<0.001	<0.001
*Uncultured*	*Unidentified*	0.22 (0.03)	<0.001	<0.001
*Unidentified*	*Unidentified*	0.21 (0.05)	<0.001	<0.001
*Ruminococcaceae*	*Ruminococcaceae UCG-009*	0.21 (0.05)	<0.001	<0.001
*Burkholderiaceae*	*Oxalobacter*	0.21 (0.06)	<0.001	0.001
*Lachnospiraceae*	*Lactonifactor*	0.20 (0.06)	0.001	0.001
*Ruminococcaceae*	*CAG 352*	0.20 (0.05)	<0.001	0.001
*Christensenellaceae*	*Unidentified*	0.20 (0.04)	<0.001	<0.001
*Lachnospiraceae*	*GCA 900066575*	0.20 (0.05)	<0.001	0.001
*Eubacteriaceae*	*Eubacterium*	0.19 (0.05)	<0.001	0.001
*Eggerthellaceae*	*Uncultured*	0.19 (0.04)	<0.001	<0.001
*Eggerthellaceae*	*Gordonibacter*	0.19 (0.04)	<0.001	<0.001
*Erysipelotrichaceae*	*Solobacterium*	0.17 (0.04)	<0.001	<0.001
*Rikenellaceae*	*Rikenellaceae RC9 gut group*	0.16 (0.05)	0.001	0.001
*Uncultured*	*Gut metagenome*	0.16 (0.04)	0.001	0.001
*Saccharimonadaceae*	*Uncultured*	0.16 (0.05)	0.001	0.001
*Lachnospiraceae*	*Lachnospiraceae UCG-010*	0.15 (0.04)	0.001	0.001
*Veillonellaceae*	*Megasphaera*	0.14 (0.04)	0.001	0.001

Note: Taxa were adjusted for age, sex, body mass index, diet quality, and irritable bowel syndrome at baseline and for multiple comparisons using Benjamani–Hochberg correction (*q* < 0.05).

**Table 4 microorganisms-12-01960-t004:** Adjusted analyses of the change in differential abundance of MetaCyc pathways.

Pathway Name	Pathway Description	Coefficient (SE)	*p*-Value	*q*-Value
Ethylmalonyl-CoA pathway (PWY.5741)	Allows certain bacteria to metabolise simple carbon (C1) compounds (e.g., methanol, carbon dioxide) into key intermediates for cellular growth and biosynthesis.	0.43 (0.08)	<0.001	<0.001
Sucrose degradation II [sucrose synthase] (PWY.3801)	An alternative pathway in which sucrose is cleaved into glucose and fructose, and then further metabolised to provide energy and carbon for cellular processes.	0.41 (0.06)	<0.001	<0.001
Superpathway of lipopolysaccharide biosynthesis (LPSSYN.PWY)	The set of biochemical processes used by Gram-negative bacteria to produce lipopolysaccharides.	0.39 (0.10)	<0.001	<0.001
Methylaspartate cycle (PWY.6728)	A metabolic pathway used by some microorganisms to assimilate C3 carbon compounds, particularly propionate (a three-carbon short-chain fatty acid), as a carbon and energy source.	0.38 (0.09)	<0.001	<0.001
Vitamin E biosynthesis [tocopherols] (PWY.1422)	The metabolic process by which certain bacteria synthesise tocopherols, the active form of Vitamin E, an antioxidant.	0.37 (0.05)	<0.001	<0.001
Mycolyl–arabinogalactan–peptidoglycan complex biosynthesis (PWY.6397)	The pathway responsible for the production of the mycolyl–arabinogalactan–peptidoglycan complex, which is a major component of the mycobacterial cell wall.	0.37 (0.05)	<0.001	<0.001
Starch degradation III (PWY.6731)	A biochemical process by which starch is broken down into simpler sugars that can be utilised for energy and other metabolic functions.	0.30 (0.06)	<0.001	<0.001
Protein N-glycosylation [bacterial] (PWY.7031)	The biochemical process in bacteria where proteins are modified by the attachment of glycans (sugar chains) to specific nitrogen atoms on asparagine residues in proteins, which can play a key role in protein stability, pathogenicity, immune evasion, and biofilm formation.	0.29 (0.07)	<0.001	<0.001
S-methyl-5-thio-α-D-ribose 1-phosphate degradation I (PWY.4361)	A pathway involved in the breakdown of a compound called S-methyl-5-thio-α-D-ribose 1-phosphate (MTR-1P), part of a broader process called methionine salvage, where sulphur-containing molecules are recycled for use in the synthesis of methionine.	0.29 (0.06)	<0.001	<0.001
L-methionine salvage cycle III (PWY.7527)	A pathway where L-methionine is regenerated from methylthioadenosine (MTA), allowing for the recycling of sulphur back into methionine.	0.28 (0.07)	<0.001	<0.001

Note: Analyses were further adjusted for age, sex, body mass index, diet quality, and irritable bowel syndrome at baseline and for multiple comparisons using Benjamani–Hochberg correction (*q* < 0.05).

**Table 5 microorganisms-12-01960-t005:** Associations between the change in depressive symptom scores and change in the relative abundance of taxa at the genus level one month after bowel preparation and colonoscopy compared to baseline.

Genus	Coefficient (SE)	*p*-Value	*q*-Value
Negatively associated			
*Ruminococcaceae UCG-009*	−0.94 (0.23)	<0.001	0.028
*Harryflintia*	−0.57 (0.19)	0.004	0.215
*Unidentified*	−0.36 (0.17)	0.036	0.841
*Uncultured*	−0.23 (0.08)	0.005	0.215
*Klebsiella*	−0.14 (0.05)	0.006	0.230
Positively associated			
*Uncultured*	0.15 (0.04)	0.001	0.103
*Haemophilus*	0.13 (0.06)	0.044	0.841
*Granulicatella*	0.22 (0.10)	0.036	0.841

Notes: Linear regression analyses were adjusted for age, sex, body mass index, diet quality, irritable bowel syndrome at baseline, and for multiple comparisons using Benjamani–Hochberg correction (*q* < 0.05).

## Data Availability

The original contributions presented in the study are included in the article/Appendix A, further inquiries can be directed to the corresponding author.

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
