# Peer review of "Depressive Symptoms and Gut Microbiota after Bowel Preparation and Colonoscopy: A Pre–Post Intervention Study"

_microorganisms, 2024, doi:10.3390/microorganisms12101960_

Round 1

Reviewer 1 Report

Comments and Suggestions for Authors

microorganisms-3189428

Depressive symptoms and gut microbiota after bowel preparation and colonoscopy: a pre-post intervention study

The manuscript by McGuinness et al. described a pre-post intervention study in adults one week before and one month after mechanical bowel preparation and colonoscopy. The authors examined changes in average anxiety, stress, quality of life scores, and gut microbiota composition using 16S rRNA sequencing. Overall, the manuscript was appropriately prepared, and the data were sufficient for the conclusion. However, there are several issues to consider as follows.

1. Figure 2: Please increase the size of texts and numbers.

2. Tables S3 and S5 can be moved to the main text as they are essential results.

3. Since the figure legends are available, please omit the figure title created by R (Figure 2a, 2b, 3).

4. Please discuss the differential pathways (table S4) to identify whether they have any biological meaning.

Author Response

  1. Figure 2: Please increase the size of texts and numbers.

Thank you for this feedback. We have increased the sizes as requested.

  1. Tables S3 and S5 can be moved to the main text as they are essential results.

Thank you for this feedback. We have moved Tables S3-S5 to the main results as we agree that these tables comprised main findings.

  1. Since the figure legends are available, please omit the figure title created by R (Figure 2a, 2b, 3).

Thank you for identifying this. We have removed the figure title created by R as requested.

  1. Please discuss the differential pathways (table S4) to identify whether they have any biological meaning.

 Thank you for this feedback. We have included the pathway description in Table 4 for clarity, and the following text in the discussion:

“We also observed higher abundances of ten MetaCyc pathways, reflecting a range of different processes relating to nutrient metabolism, one-month post-procedure. However, as this study used 16S rRNA sequencing we could only measure inferred functionality using PICRUSt2, therefore the biological relevance is unclear and requires exploration with shotgun metagenomic sequencing.”

Reviewer 2 Report

Comments and Suggestions for Authors

Review of the manuscript microorganisms-3189428

Depressive symptoms and gut microbiota after bowel prepara-2 tion and colonoscopy: a pre-post intervention study

By Amelia J. McGuinness et al.

The evaluated manuscript is an original article. The authors aimed to investigate the influence of mechanical bowel preparation (MBP) applied prior to colonoscopy on the quantity and composition of intestinal microbiota, as well as on depressive symptoms. The analysis was conducted comparatively, assessing the parameters one week before and one month after the colonoscopy. The authors demonstrated a decrease in average depressive scores and an increase in average quality of life scores one month post-procedure compared to baseline. Furthermore, the changes in average depression scores appeared to be moderated by coexisting irritable bowel syndrome (IBS), with individuals suffering from IBS experiencing a worsening of their depressive symptoms, while those without IBS reported improvements. Compared to baseline, the authors also observed a reduction in gut microbiota alpha diversity, modest effects on beta diversity, and differentially abundant genera and functional pathways. Finally, increases in the genus Ruminococcaceae UCG-009 were noted and associated with improvements in depressive symptoms.

Reading the manuscript raises several questions and concerns.

Section 2.2: Participants and Settings, and Table 1. The patients included in the study should be described in greater detail regarding their clinical characteristics. It is well established that the microbiota is closely influenced by both physiological and pathophysiological conditions. Consequently, various chronic diseases—particularly chronic kidney disease and diabetes and others—as well as lifestyle factors such as smoking, alcohol consumption, and the use of energy drinks and other stimulants, can significantly impact the microbiota. This should be clarified.

Subchapter 2.2: Participants and Settings Among the exclusions from the study, there is information regarding the antibiotics used by the patients (There were no exclusions regarding antibiotic use). The paper does not specify how it was ensured that the patients included in the study had not used any antibiotics prior to the study, as this could significantly impact the initial microbiota assessment. Was this information obtained through a medical interview, a review of medications, or self-reporting by the patients?

2.3. Intervention. Why were stool samples self-collected by patients one week prior to their colonoscopy examination stored at home at -20°C instead of being immediately delivered to the study site and stored at -80°C, as was done with the samples collected directly during the colonoscopy? Could the difference in storage temperature affect the subsequent sequencing of the 16S rRNA gene?

2.4. Intervention. Patients were instructed to begin a low-fiber ( diet two days before their colonoscopy. They were then required to fast for 12 to 24 hours prior to the procedure and to consume a sodium picosulfate-based bowel preparation product in three separate doses. How did the patients tolerate the contact laxative? Abdominal cramps and diarrhea are known common side effects of picosulfate. Did these potential side effects impact the patients' baseline self-reports of depression and well-being?

2.5. Outcomes. The authors utilized the Hospital Anxiety and Depression Scale (HADS) to evaluate depression, although there are several other widely recognized instruments available, such as the Beck Depression Inventory (BDI), which is endorsed by NICE, the Center for Epidemiological Studies Depression Scale (CES-D), or the Montgomery-Åsberg Depression Rating Scale (MADRS). Why HADS have been chosen? In the authors' opinion, does this scale offer any advantages over other diagnostic tools of a similar nature? Do the authors possess the most experience in utilizing this scale, among other considerations?

3.1. Results. What accounts for the significant loss of participants from the study? Out of the initially recruited 136 individuals, only 59 completed the study.

3.1. Results. Additionally, what specific antidepressants and antireflux medications were administered to the patients?

3.1. Results. I do not understand why patients treated with antidepressants (11 out of 59) were analyzed alongside the other participants. The initial self-assessment of depression, as well as the assessment following colonoscopy, is likely different for these patients compared to those who did not receive antidepressant treatment. In my opinion, this represents a significant methodological concern in the study design.

3.1. Results and Discussion. The patient characteristics indicate that individuals included in the study—both those who completed it and those lost to follow-up—were nearly obese, with a mean BMI of 29.7. The discussion lacks a cohesive examination of the potential impact of this factor on the microbiota, despite the well-established understanding that overweight and obesity significantly influence microbiota composition (e.g. https://www.nature.com/articles/s41575-023-00867-z#citeas, https://www.ncbi.nlm.nih.gov/pmc/articles/PMC8291023/ etc.).

Author Response

Section 2.2: Participants and Settings, and Table 1. The patients included in the study should be described in greater detail regarding their clinical characteristics. It is well established that the microbiota is closely influenced by both physiological and pathophysiological conditions. Consequently, various chronic diseases—particularly chronic kidney disease and diabetes and others—as well as lifestyle factors such as smoking, alcohol consumption, and the use of energy drinks and other stimulants, can significantly impact the microbiota. This should be clarified.

Thank you for highlighting this limitation. Our study was observational and recruited from a convenience sample of adults undergoing colonoscopy from our local public hospital. Consequently, they represent a heterogenous, although possibly more representative, sample. We have therefore included the following in our limitations section:

“As we recruited a convenience sample of adults undergoing colonoscopy at our regional public hospital, our study population was highly heterogenous in terms of their baseline clinical characteristics, and the impact of these differences on baseline gut microbiota composition and reestablishment are unclear.”

Subchapter 2.2: Participants and Settings Among the exclusions from the study, there is information regarding the antibiotics used by the patients (There were no exclusions regarding antibiotic use). The paper does not specify how it was ensured that the patients included in the study had not used any antibiotics prior to the study, as this could significantly impact the initial microbiota assessment. Was this information obtained through a medical interview, a review of medications, or self-reporting by the patients?

Thank you for this feedback. The original protocol for this study, developed in 2015/2016 when gut microbiome research was largely still in its nascency, did not have the use of antibiotics as an exclusion criterion. However, as part of their baseline questionnaires (one week before colonoscopy) participants were required to self-report any medications taken, and no participants reported the use of antibiotics. We have amended the typographical error regarding this in the results section, and included the following in the limitations section:

“Whilst no participants reported antibiotic use within one week of colonoscopy, the use of antibiotics within one month of colonoscopy was not an exclusion criterion, which may have impacted baseline composition.”

2.3. Intervention. Why were stool samples self-collected by patients one week prior to their colonoscopy examination stored at home at -20°C instead of being immediately delivered to the study site and stored at -80°C, as was done with the samples collected directly during the colonoscopy? Could the difference in storage temperature affect the subsequent sequencing of the 16S rRNA gene?

Both the pre- and post-colonoscopy samples were stored in a home freezer, and then transferred on ice to a study researcher within a week for storage at -80 degrees. As this was an unfunded study, this choice was to reduce the burden of participant visits to the study centre, and instead align their returning of stool with their date of colonoscopy (sample 1) and their outpatient follow up (sample 3). As the intra-operative sample was not statistically compared to the others, the impact of faster storage at -80 degrees is unlikely to have impacted the present results. However, we have included the following in the limitations section:

“Stool samples were collected at home and stored in participant home freezers for one week prior to transfer to long-term storage, and the impact of the delay in storage at −80℃ is also unclear.”

2.4. Intervention. Patients were instructed to begin a low-fiber ( diet two days before their colonoscopy. They were then required to fast for 12 to 24 hours prior to the procedure and to consume a sodium picosulfate-based bowel preparation product in three separate doses. How did the patients tolerate the contact laxative? Abdominal cramps and diarrhea are known common side effects of picosulfate. Did these potential side effects impact the patients' baseline self-reports of depression and well-being?

Thank you for this question. Participants were asked to complete their baseline questionnaires, including those pertaining to mental health, one week prior to commencing the bowel preparation processes. As such, these baseline measures were collected pre-intervention (i.e., before both bowel preparation and colonoscopy), thus the gastrointestinal side effects induced by the bowel preparation will not have impacted on our baseline psychological outcomes.

2.5. Outcomes. The authors utilized the Hospital Anxiety and Depression Scale (HADS) to evaluate depression, although there are several other widely recognized instruments available, such as the Beck Depression Inventory (BDI), which is endorsed by NICE, the Center for Epidemiological Studies Depression Scale (CES-D), or the Montgomery-Åsberg Depression Rating Scale (MADRS). Why HADS have been chosen? In the authors' opinion, does this scale offer any advantages over other diagnostic tools of a similar nature? Do the authors possess the most experience in utilizing this scale, among other considerations?

Thank you for this question. We selected the HADS as, unlike the other mentioned instruments, it has been specifically developed and validated to assess symptoms of depression and anxiety in the setting of a hospital medical outpatient clinic (1, 2), which is the setting from which our study participants were recruited. Other advantages of this tool are its short length (14 questions compared to 20 and 21 for the CES-D and BDI, respectively) which is important for reducing participant burden, as well as it being a self-reported measure, reducing the need for a clinical appointment, such as is required for the MADRS. Together, the specific design of this tool for our study population, in addition with the benefits that reduced participant burden, drove our choice for this tool.

  1. Zigmond, A. S., & Snaith, R. P. (1983). The Hospital Anxiety and Depression Scale. Acta Psychiatrica Scandinavica, 67(6), 361-370. https://doi.org/10.1111/j.1600-0447.1983.tb09716.x
  2. Herrmann, C. (1997). International experiences with the Hospital Anxiety and Depression Scale-a review of validation data and clinical results. Journal of Psychosomatic Research, 42(1), 17-41. https://doi.org/10.1016/S0022-3999(96)00216-4

3.1. Results. What accounts for the significant loss of participants from the study? Out of the initially recruited 136 individuals, only 59 completed the study.

Thank you for drawing our attention to this crucial information. We have amended Section 3.1. and the accompanying flow diagram to highlight where loss of participants occurred and why. Briefly, of the initial 136 participants that provided informed consent, 45 participants did not provide any data towards the study for varying reasons: cancelled colonoscopy (n=1), didn’t notify study that their colonoscopy had been booked in/completed or had an unexpected data change (n=9), didn’t collect a baseline faecal sample (n=2), no longer wished to participate (n=2), no questionnaire data (n=2), withdrew with no reason provided (n=12), did not respond to communication with study staff (n=4), study was completed before participant received their colonoscopy (n=5), too busy/not enough time/travel commitments (n=5), study requirements were too much (n=2), unable to take faecal sample (n=3), or unwell (n=3). Of the remaining 86 participants that provided baseline data, 27 did not complete the study for the following reasons: unable to be contacted and considered lost to follow up (n=20), cancer diagnosis (n=2), moved away (n=1), personal reasons (n=1), study requirements were too much (n=3). This left 59 participants for the final complete case analyses.

3.1. Results. Additionally, what specific antidepressants and antireflux medications were administered to the patients?

Thank you for this question. As this was an observational study, antidepressant and anti-reflux medications were not specifically administered to patients. Rather, some participants were taking these at baseline. We have updated Table 1 to list the specific antidepressant and antireflux medications self-reported by participants.

3.1. Results. I do not understand why patients treated with antidepressants (11 out of 59) were analyzed alongside the other participants. The initial self-assessment of depression, as well as the assessment following colonoscopy, is likely different for these patients compared to those who did not receive antidepressant treatment. In my opinion, this represents a significant methodological concern in the study design.

Thank you for this feedback. In response to your query, we have included an additional post-hoc exploratory analysis exploring whether the change in HADS depression scores were moderated by baseline self-reported anti-depressant use. We did not observe evidence of a statistically significant interaction, supporting our choice to analyse all participants together.

3.1. Results and Discussion. The patient characteristics indicate that individuals included in the study—both those who completed it and those lost to follow-up—were nearly obese, with a mean BMI of 29.7. The discussion lacks a cohesive examination of the potential impact of this factor on the microbiota, despite the well-established understanding that overweight and obesity significantly influence microbiota composition (e.g. https://www.nature.com/articles/s41575-023-00867-z#citeas, https://www.ncbi.nlm.nih.gov/pmc/articles/PMC8291023/ etc.).

Thank you for this comment. In response to your query, we have included an additional post-hoc exploratory analysis exploring whether the change in HADS depression scores were moderated by baseline BMI, dichotomised as above or below 30. We observed evidence of a statistically significant two-way interaction; Those with a BMI equal to or greater than 30 appeared to have reductions in HADS depression scores, whereas those with a BMI below 30 appeared to have minimal change in HADS depression scores, one-month post-procedure compared to baseline. We have also included a section about this in the discussion, as requested.

Round 2

Reviewer 2 Report

Comments and Suggestions for Authors

Thank you for providing the revised version of the manuscript. The authors have responded to my comments and have taken into account my suggestions. I believe that the manuscript can be published.